# Kernelised Normalising Flows

**Eshant English**[*1]  **Matthias Kirchler**[*1,2]  **Christoph Lippert**[1,3]

`{first}.{last}@hpi.de`

[1]Hasso Plattner Institute for Digital Engineering, Germany
[2]University of Kaiserslautern-Landau, Germany
[3]Hasso Plattner Institute for Digital Health at the Icahn School of Medicine at Mount Sinai, NYC, USA

## Abstract

Normalising Flows are non-parametric statistical models characterised by their dual capabilities of density estimation and generation. This duality requires an inherently invertible architecture. However, the requirement of invertibility imposes constraints on their expressiveness, necessitating a large number of parameters and innovative architectural designs to achieve good results. Whilst flow-based models predominantly rely on neural-network-based transformations for expressive designs, alternative transformation methods have received limited attention. In this work, we present Ferumal flow, a novel kernelised normalising flow paradigm that integrates kernels into the framework. Our results demonstrate that a kernelised flow can yield competitive or superior results compared to neural network-based flows whilst maintaining parameter efficiency. Kernelised flows excel especially in the low-data regime, enabling flexible non-parametric density estimation in applications with sparse data availability.

## 1 Introduction

Maximum likelihood is a fundamental approach to parameter estimation in the field of machine learning and statistics. However, its direct application to deep generative modelling is rare due to the intractability of the likelihood function. Popular probabilistic generative models such as Diffusion Models (Dai & Seljak, 2020b) and Variational Autoencoders (Kingma & Welling, 2022) instead resort to optimising the Evidence Lower Bound (ELBO), a lower bound on the log-likelihood, due to the challenges in evaluating likelihoods.

The change of variables theorem offers a neat and elegant solution to compute the exact likelihood for deep generative modelling. These models, known as normalising flows, employ invertible architectures to transform complex probability distributions into simpler ones. Normalising flows (Papamakarios et al., 2021; Kobyzev et al., 2021) excel in efficient density estimation and exact sampling, making them suitable for various applications.

Whilst flow-based models possess appealing properties rooted in invertibility, they also impose limitations on modelling choices, which can restrict their expressiveness. This limitation can be mitigated by employing deeper models with a higher number of parameters. For instance, the Glow model Kingma & Dhariwal (2018) utilised approximately 45 million parameters for image generation on CIFAR-10 Krizhevsky (2009a), whereas StyleGAN3 Karras et al. (2019), a method that doesn't use likelihood optimisation, achieved a superior FID score with only about a million parameters.

The issue of over-parameterisation in flow-based models hinders their effectiveness in domains with limited data, such as medical applications. For example, normalising flows can be used to model complex phenotypic or genotypic data in genetic association studies (Hansen et al., 2022; Kirchler et al., 2022); collection of high-quality data in these settings is expensive, with many studies only including data on a few hundred to a few thousand instances. In scenarios with low data availability, a flow-based network can easily memorise the entire dataset, leading to an unsatisfactory performance on the test set. While existing research has focused on enhancing the expressiveness of flows through clever architectural techniques, the challenge of achieving parameter efficiency has mostly been overlooked with few exceptions Lee et al. (2020). Most normalising flows developed to date rely on

---

[*]Equal contribution

neural networks to transform complex distributions into simpler distributions. However, there is no inherent requirement for flows to use neural networks. Due to their over-parameterisation and low inductive bias, neural networks tend to struggle with generalisation in the low-data regime, making them inapplicable to many real-world applications.

In this work, we propose a novel approach to flow-based distribution modelling by replacing neural networks with kernels. Kernel machines work well in low-data regimes and retain expressiveness even at scale. We introduce Ferumal flows, a kernelised normalising flow paradigm that outperforms or achieves competitive performance in density estimation for tabular data compared to other efficiently invertible neural network baselines like RealNVP and Glow. Ferumal flows require up to 93% fewer parameters than their neural network-based counterparts whilst still matching or outperforming them in terms of likelihood estimation. We also investigate efficient training strategies for larger-scale datasets and show that kernelising the flows works especially well on small datasets.

## 2 BACKGROUND

### 2.1 MAXIMUM LIKELIHOOD OPTIMISATION WITH NORMALISING FLOWS

A normalising flow is an invertible neural network, $f : \mathbb{R}^D \to \mathbb{R}^D$ that maps real data $x$ onto noise variables $z$. The noise variable $z$ is commonly modelled by a simple distribution with explicitly known density, such as a normal or uniform distribution, while the distribution of $x$ is unknown and needs to be estimated. As normalising flows are maximum likelihood estimators, for a given data set of instances $x_1, \ldots, x_n$, we want to maximise the log-likelihood over model parameters,

$$\max_f \sum_{i=1}^n \log \left( p_X(x_i) \right).$$

With the change of variables formula,

$$p_X(x) = p_Z \left( f(x) \right) \left| \det J_f(x) \right|,$$

where $J_f(x)$ is the Jacobian of $f$ in $x$, we can perform this optimisation directly:

$$\max_f \sum_{i=1}^n \log \left( p_Z \left( f\left(x_i\right) \right) \right) + \log \left( \left| \det J_f\left(x_i\right) \right| \right).$$

The first part, $\log(p_Z(f(x_i)))$, can be computed in closed form due to the choice of $p_Z$. The second part, $\log(|\det J_f(x_i)|)$, can only be computed efficiently if $f$ is designed appropriately.

### 2.2 COUPLING LAYERS

One standard way to design such an invertible neural network $f$ with tractable Jacobian is affine coupling layers (Dinh et al., 2017). A coupling layer $C_\ell : \mathbb{R}^D \to \mathbb{R}^D$ is an invertible layer that maps an input $y_{\ell-1}$ to an output $y_\ell$ ($\ell$ is the layer index): first, permute data dimensions with a fixed permutation $\pi_\ell$, and split the output into the first $d$ and second $D - d$ dimensions:

$$\left[ u_\ell^1, \ u_\ell^2 \right] = \left[ \pi_\ell(y_{\ell-1})_{1:d}, \ \pi_\ell(y_{\ell-1})_{d+1:D} \right].$$

Commonly, the permutation is either a reversal of dimensions or picked uniformly at random. Next, we apply two neural networks, $s_\ell, t_\ell : \mathbb{R}^d \to \mathbb{R}^{D-d}$, to the first output, $u_\ell^1$, and use it to scale and translate the second part, $u_\ell^2$:

$$y_\ell^2 = \exp \left( s_\ell \left( u_\ell^1 \right) \right) \odot u_\ell^2 + t_\ell \left( u_\ell^1 \right).$$

The first part of $y_\ell$ remains unchanged, i.e., $y_\ell^1 = u_\ell^1$, and we get the output of the coupling layer as:

$$y_\ell = C_\ell \left( y_{\ell-1} \right) = \left[ y_\ell^1, \ y_\ell^2 \right] = \left[ u_\ell^1, \ \exp \left( s_\ell \left( u_\ell^1 \right) \right) \odot u_\ell^2 + t_\ell \left( u_\ell^1 \right) \right].$$

The Jacobian matrix of this transformation is a permutation of a lower triangular matrix resulting from $u_\ell^1$ undergoing an identity transformation and $u_\ell^2$ getting transformed elementwise by a function of $u_\ell^1$. The Jacobian of the permutation has a determinant with an absolute value of 1 by default.

The diagonal of the remaining Jacobian consists of $d$ elements equal to unity and the other $D - d$ elements equal the scaling vector $s_\ell\left(u_\ell^1\right)$. Thus, the determinant can be efficiently computed as the product of the elements of the scaling vector $s_\ell\left(u_\ell^1\right)$.

The coupling layers are also efficiently invertible as only some of the dimensions are transformed and the unchanged dimensions can be used to obtain the scaling and translation factors used for the forward transformation to reverse the operation.

Multiple coupling layers can be linked in a chain of $L$ layers such that all dimensions can be transformed:

$$f(x) = C_L \circ \cdots \circ C_1(x),$$

i.e., $y_0 = x$ and $y_L = f(x)$.

## 2.3 KERNEL MACHINES

Kernel machines (Schölkopf et al., 2002) implicitly map a data vector $u \in \mathbb{R}^p$ into a high-dimensional (potentially infinite-dimensional) reproducing kernel Hilbert space (RKHS), $\mathcal{H}$, by means of a fixed feature map $\phi : \mathbb{R}^p \to \mathcal{H}$. The RKHS $\mathcal{H}$ has an associated positive definite kernel $k(u, v) = \langle \phi(u), \phi(v) \rangle_{\mathcal{H}}$, where $\langle \cdot, \cdot \rangle_{\mathcal{H}} : \mathcal{H} \times \mathcal{H} \to \mathbb{R}$ is the inner product of $\mathcal{H}$. The kernel $k$ can oftentimes be computed in closed form without requiring the explicit mapping of $u, v$ into $\mathcal{H}$, making computation of many otherwise intractable problems feasible. In particular, as many linear learning algorithms, such as Ridge Regression or Support Vector Machines, only require explicit computations of norms and inner products, these algorithms can be efficiently kernelised. Instead of solving the original learning problem in $\mathbb{R}^p$, kernel machines map data into the RKHS and solve the problem with a linear algorithm in the RKHS, offering both computational efficiency (due to linearity and kernelisation) and expressivity (due to the nonlinearity of the feature map and the high dimensionality of the RKHS).

## 3 FERUMAL FLOWS: KERNELISATION OF FLOW-BASED ARCHITECTURES

In this section, we extend standard coupling layers to use kernel-based scaling and translation functions instead. Whilst neural networks are known to perform well in large-data regimes or when transfer-learning from larger datasets can be applied, kernel machines perform well even at small sample sizes and naturally trade-off model complexity against dataset size without losing expressivity.

## 3.1 KERNELISED COUPLING LAYERS

We keep the definition of coupling layers in Section 2, and only replace the functions $s_\ell$ and $t_\ell$ by functions mapping to and from an RKHS $\mathcal{H}$. We have to deal with two main differences to the kernelisation of many other learning algorithms: firstly, the explicit likelihood optimisation does not include a regularisation term that penalises the norm of the prediction function. And secondly, instead of a single mapping from origin space to RKHS to single-dimensional output, we aim to combine multiple layers, in each of which the scaling and translation map from origin space to RKHS to multi-dimensional output. As a result, the optimisation problem will not be convex in contrast to standard kernel learning, and we have to derive a kernelised and tractable representation of the learning objective.

In particular, in layer $\ell$, we introduce RKHS elements $V_\ell^s, V_\ell^t \in \mathcal{H}^{D-d}$ and define scaling and translation as

$$s_\ell\left(u_\ell^1\right) = \left[\left\langle V_{\ell,j}^s, \phi\left(u_\ell^1\right)\right\rangle_{\mathcal{H}}\right]_{j=1}^{D-d} \in \mathbb{R}^{D-d} \qquad \text{and} \qquad t_\ell\left(u_\ell^1\right) = \left[\left\langle V_{\ell,j}^t, \phi\left(u_\ell^1\right)\right\rangle_{\mathcal{H}}\right]_{j=1}^{D-d} \in \mathbb{R}^{D-d}.$$

We summarise $V_\ell = [V_\ell^s, V_\ell^t]$ and $V = [V_1, \ldots, V_L]$ for the full flow $f(x) = C_L \circ \cdots \circ C_1(x)$. Since elements $V \in \mathcal{H}^{2L(D-d)}$ and $\mathcal{H}$'s dimensionality is potentially infinite, we cannot directly optimise the objective:

$$\max_V \sum_{i=1}^n p_Z\left(C_L \circ \ldots \circ C_1(x_i)\right) + \log\left(\left|\det J_{C_L \circ \ldots \circ C_1}(x_i)\right|\right) = L(V). \tag{1}$$

However, we can state a version of the representer theorem (Schölkopf et al., 2001) that allows us to kernelise the objective:

**Proposition 3.1.** *Given the objective $L$ in equation* (1)*, for any $V' = [V'_1, \ldots, V'_L] \in \mathcal{H}^{L(D-d)}$ there also exists a $V$ with $L(V) = L(V')$ such that*

$$V_\ell = \sum_{i=1}^{n} k\left(\cdot, u^1_{\ell,i}\right) A_{\ell,i}$$

*for some $A_{\ell,i} = \left[A^s_{\ell,i}, A^t_{\ell,i}\right] \in \mathbb{R}^{2(D-d)}$. Here, $u^1_{\ell,i} = \pi_\ell(C_{\ell-1} \circ \cdots \circ C_1(x_i))_{1:d}$, i.e., the first part of the permutated input to layer $\ell$ for data point $i$. In particular, if there exists a solution $V' \in \arg\max_V L(V)$, then there's also a solution $V^*$ of the form*

$$V^*_\ell = \sum_{i=1}^{n} k\left(\cdot, u^1_{\ell,i}\right) A_{\ell,i}.$$

*Proof.* Let $V' \in \mathcal{H}^{2L(D-d)}$ and $\Phi_\ell = \text{span}\{\phi(u^1_{\ell,1}), \ldots, \phi(u^1_{\ell,n})\}$ be the space spanned by the feature maps of layer $\ell$ inputs, and let $\Phi_\ell^\perp$ denote its orthogonal complement in $\mathcal{H}$. We can then represent each element $V^s_{\ell,j}{}' \in \mathcal{H}$ ($j \in \{1, \ldots, D-d\}$) as an orthogonal sum of an element of $\Phi_\ell$ and $\Phi_\ell^\perp$,

$$V^s_{\ell,j}{}' = \phi_{\ell,j} + \phi^\perp_{\ell,j}, \text{ with } \phi_{\ell,j} = \sum_{i=1}^{n} A^s_{\ell,i,j} \phi(u^1_{\ell,i}) \text{ and } \langle \phi^\perp_{\ell,j}, \phi(u^1_{\ell,i}) \rangle_\mathcal{H} = 0 \; \forall i = 1, \ldots, n$$

for some values $A^s_{\ell,i,j} \in \mathbb{R}$. In the objective (1), we only use $V^s_{\ell,j}{}'$ to compute $\langle V^s_{\ell,j}{}', \phi(u^1_{\ell,i}) \rangle_\mathcal{H}$ as part of the computation of $s_\ell(u_{\ell,i})$. But due to orthogonality, it holds that

$$\langle V^s_{\ell,j}{}', \phi(u^1_{\ell,i}) \rangle_\mathcal{H} = \langle \phi_{\ell,j} + \phi^\perp_{\ell,j}, \phi(u^1_{\ell,i}) \rangle_\mathcal{H} = \langle \phi_{\ell,j}, \phi(u^1_{\ell,i}) \rangle_\mathcal{H} + \langle \phi^\perp_{\ell,j}, \phi(u^1_{\ell,i}) \rangle_\mathcal{H} = \langle \phi_{\ell,j}, \phi(u^1_{\ell,i}) \rangle_\mathcal{H}.$$

Hence, replacing $V^s_{\ell,j}{}'$ by $\phi_{\ell,j} = \sum_{i=1}^{n} A^s_{l,i,j} \phi(u^1_{\ell,i})$ keeps the objective unchanged. The reproducing property of the RKHS $\mathcal{H}$ now states that $\langle \phi(u^1_{\ell,i}), \phi(\cdot) \rangle_\mathcal{H} = k(u^1_{\ell,i}, \cdot)$.

Repeating this for all $\ell = 1, \ldots, L$, all $j = 1, \ldots, D-d$ and also for translations $t_\ell$ yields a version of $V'$ that can be represented as a linear combination of the stated form. $\square$

In contrast to the classical represener theorem, the objective doesn't contain a regulariser of the model's norm, which would ensure that *any* solution can necessarily be represented as a linear combination of kernel evaluations. However, if a solution exists, Proposition 3.1 ensures that there also exists a solution that can be expressed as a linear combination of kernel evaluations. Therefore, we can re-insert this solution $V^*$ into the objective 1 to get a kernelised objective.

For layer $\ell$ and arbitrary $a \in \mathbb{R}^d$,

$$s_\ell(a) = \left[\langle V^{s*}_{\ell,j}, \phi(a) \rangle\right]_{j=1}^{D-d} = \left[\sum_{i=1}^{n} A^s_{\ell,i,j} k\left(u^1_{\ell,i}, a\right)\right]_{j=1}^{D-d} = \sum_{i=1}^{n} k\left(u^1_{\ell,i}, a\right) A^s_{\ell,i}.$$

As in the objective (1), $s_\ell$ gets only evaluated in points $a \in \left\{u^1_{\ell,i} | i = 1, \ldots, n\right\}$, this simplifies to

$$s_\ell\left(u^1_{\ell,m}\right) = \sum_{i=1}^{n} k\left(u^1_{\ell,i}, u^1_{\ell,m}\right) A^s_{\ell,i} = A^s_\ell K\left(U^1_\ell, U^1_\ell\right)_m,$$

where $K\left(U^1_\ell, U^1_\ell\right) = \left[k\left(u^1_{\ell,i}, u_{\ell,m}\right)^1\right]_{i,m=1}^{n}$ is the kernel matrix at layer $\ell$ and $A^s_\ell = \left[A^s_{\ell,1}, \ldots, A^s_{\ell,n}\right] \in \mathbb{R}^{(D-d) \times n}$ is the weight matrix. A similar derivation holds for $t_\ell$.

In total, we can kernelise the objective (1) and optimise over parameters $A \in \mathbb{R}^{L \times n \times 2(D-d)}$ instead of over $V \in \mathcal{H}^{2L(D-d)}$. In contrast to neural network-based learning, the number of parameters, $2Ln(D-d)$, is fixed except for the number of layers (since $d$ is usually set to $\lfloor D/2 \rfloor$), but increases linearly with the dataset size, $n$. This makes kernelised flows especially promising for learning

in the low-data regime, as their model complexity naturally scales with dataset size and does not over-parametrise as much as neural networks (as long as one does not employ an excessive number of layers).

Since the resulting objective function is not convex, optimisers targeted to standard kernel machines such as Sequential Minimal Optimisation (Platt, 1998) are not applicable. Instead, we optimise (1) with variations of stochastic gradient descent (Kingma & Ba, 2014).

## 3.2 Efficient learning with auxiliary points

The basic kernelised formulation can be very computationally expensive even at moderate dataset sizes and can tend towards overfitting in the lower-data regime. In Gaussian Process (GP) regression, this problem is usually addressed via sparse GPs and the introduction of *inducing variables* (Quiñonero-Candela & Rasmussen, 2005). In a similar spirit, we introduce *auxiliary points*. In layer $\ell$, instead of computing the kernel with respect to the full data $u^1_{\ell,1}, \ldots, u^1_{\ell,n}$, we take $N \ll n$ new variables $W^1_\ell = [w^1_{\ell,1}, \ldots, w^1_{\ell,N}] \in \mathbb{R}^{d \times N}$ and compute the scaling transform as

$$\hat{s}_\ell(u^1_{\ell,m}) = \hat{A}^s_\ell K(U^1_\ell, W^1_\ell)_m$$

with $\hat{A}^s_\ell \in \mathbb{R}^{(D-d) \times N}$ (analogously for $\hat{t}_\ell$). We make these auxiliary points learnable and initialise them to a randomly selected subset of $u^1_{\ell,1}, \ldots, u^1_{\ell,n}$. This procedure reduces the learnable parameters from $2n(D-d)L$ (for both $s_\ell$ and $t_\ell$) to $2NDL$.

In another variation we make these auxiliary points shared between layers. In particular, instead of selecting $L$ times $N$ points $w^1_{\ell,1}, \ldots, w^1_{\ell,N}$, we instead only select $W^1 = [w^1_1, \ldots, w^1_N] \in \mathbb{R}^{d \times N}$ once and compute at layer $\ell$

$$\bar{s}_\ell(u^1_{\ell,m}) = \hat{A}^s_\ell K(U^1_\ell, W^1)_m.$$

This further reduces the learnable parameters to $2N(D-d)L + 2dN$.

## 4 Related works

We are unaware of any prior work that attempts to replace neural networks with kernels in flow-based architectures directly. However, there is a family of flow models based on Iterative Gaussianisation (IG) Chen & Gopinath (2000) that utilise kernels. Notable works using Iterative Gaussianisation include Gaussianisation Flows Meng et al. (2020), Rotation-Based Iterative Gaussianisation (RBIG) Laparra et al. (2011), and Sliced Iterative Normalising Flows Dai & Seljak (2020a). These IG-based methods differ significantly from our methodology. They rely on kernel density estimation and inversion of the cumulative distribution function for each dimension individually and incorporate the dependence between input dimensions through a rotation matrix, which aims to reduce inter-dependence. In contrast, our method integrates kernels into coupling layer-based architectures. Furthermore, IG-based methods typically involve a large number of layers, resulting in inefficiency during training and a comparable number of parameters to neural network-based flow architectures. In contrast, the Ferumal flow approach of incorporating kernels can act as a drop-in replacement in many standard flow-based architectures, ensuring parameter efficiency without compromising effectiveness. Another generative model using kernels is the work on kernel transport operators (Huang et al., 2021). demonstrated promising results in low-data scenarios and favourable empirical outcomes. However, their approach differs from ours as they employed kernel mean embeddings and transfer operators, along with a pre-trained autoencoder.

Other works focusing on kernel machines in a deep learning context are deep Gaussian processes Damianou & Lawrence (2013) and deep kernel learning (Wilson et al., 2016; Wenliang et al., 2019). Deep GPs concatenate multiple layers of kernelised GP operations; however, they are Bayesian, non-invertible models for prediction tasks instead of density estimation and involve high computational complexity due to operations that require inverting a kernel matrix. Some works Rudi & Ciliberto (2021); Marteau-Ferey et al. (2021); Tsuchida et al. (2023) use kernels for nonnegative functions for modelling densities but are also not invertible. Deep kernel learning, on the other hand, designs new kernels that are parametrised by multilayer perceptrons.

Maroñas et al. (2021) integrated normalising flows within Gaussian processes. Their approach differs significantly from ours as they aimed to exploit the invertibility property of flows by applying them to the prior or the likelihood. Their combined models consist of kernels in the form of GPs but also involve neural networks in the normalising flows network, resembling more of a hybrid model.

NanoFlow Lee et al. (2020) also targets parameter efficiency in normalising flows. They rely on parameter sharing across different layers, whereas we utilise kernels. We also attempted to implement the naive parameter-sharing technique suggested by Lee et al. (2020), but we found no improvement in performance.

## 5 EXPERIMENTS

We assess the performance of our Ferumal flow kernelisation both on synthetic 2D toy datasets and on five real-world benchmark datasets sourced from Dua & Graff (2017). The benchmark datasets include Power, Gas, Hepmass, MiniBoone, and BSDS300. To ensure consistency, we adhere to the preprocessing procedure outlined by Papamakarios et al. (2018).

**Implementation details**   We kernelised the RealNVP and Glow architectures. For comparison purposes, RealNVP and Glow act as direct comparisons being the neural-net counterparts of our models. We have also included basic autoregressive methods, Masked Autoregressive Flows (Papamakarios et al., 2018), and Masked Autoregressive Distribution Estimation (Germain et al., 2015), FFJORD (Grathwohl et al., 2018) as a continuous normalising flow, and Gaussianisation Flows (Meng et al., 2020), Rotation-based Iterative Gaussianisation (Laparra et al., 2011), Sliced Iterative Normalising Flows (Dai & Seljak, 2020b) as iterative gaussianisation methods for our evaluations. Most autoregressive flow models outperform non-autoregressive flow models. However, they usually come with the trade-off of either inefficient sampling or inefficient density estimation, i.e., either the forward or the inverse computation is computationally very expensive.

**Training details**   Our Ferumal Flow kernelisation has a negligible number of hyperparameters. Apart from learning rate hyperparameters (i.e., learning rate, $\beta_1$, $\beta_2$ for Adam) and the number of layers, that are central to both kernelised and neural-net-based flows, we only need to choose a kernel with its corresponding hyperparameters (and a number of auxiliary points for large-scale experiments). This is in contrast with neural-net-based flows where choices for a flow layer include a number of hidden sublayers, the number of nodes in each sub-layer, residual connections, type of normalisation, activation function, dropout, and many more. Coupled with longer convergence times this necessitates considerably more time and resources in hyperparameter tuning than our proposed kernel methods. In our study, we utilised either the Squared Exponential kernel or Matern Kernels exclusively for all experiments. We learnt all the kernel hyperparameters using the GPyTorch library for Python for the main experiments. Throughout the experiments, we used the Adam (Kingma & Ba, 2014) optimiser, whilst adjusting the $\beta_1$



Figure 1: Histogram of 2D toy datasets. **Left:** True distribution. **Middle:** NN-based. **Right:** FF-kernelisation

and $\beta_2$ parameters of the optimiser. Additionally, we decayed the learning rate either with predefined steps (StepLR) or with cosine annealing. In all the experiments, we incorporated auxiliary points as we observed that they provided better results. In most cases, we persisted with 150 auxiliary points.

We coded our method in PyTorch (Paszke et al., 2019) and used existing implementations for the other algorithms. We ran all experiments for Ferumal flows and other baselines on CPUs (Intel Xeon 3.7 GHz). For more comprehensive training details, please refer to the Table 5

## 5.1 2D TOY DATASETS

Initially, we conducted density estimation experiments on three synthetic datasets that were sampled from two-dimensional distributions exhibiting diverse shapes and numbers of modes. Figure 1 showcases the original data distribution alongside the samples generated using the Ferumal flow kernelisation and the corresponding neural network counterpart. The neural-net-based architecture clearly shows the distortion of density in many regions whereas the kernelised

Table 1: Results on toy datasets. Log Likelihood in nats, higher is better

| Dataset | Ours (#params) | NN-based (#params) |
|---------|----------------|--------------------|
| Line | **3.75 (5K)** | 3.15 (44K) |
| Pinwheel | **-2.44 (4K)** | -2.48 (44K) |
| Moons | **-2.43 (5K)** | -2.54 (44K) |

counterpart has much better modelling. Table 1 shows the corresponding log-likelihood in nats, quantitatively showing the enhancement from our kernelisation. The results demonstrate that Ferumal flow kernelisation can outperform its neural net counterpart on these toy datasets. All the toy datasets were trained with a batch size of 200 and for 10K iterations. We also investigated the effect of our kernelisation on highly discontinuous densities strengthening our argument for kernelisation. Please refer to Appendix B

## 5.2 REAL-WORLD DATASETS

We conducted density estimation experiments on five real-world tabular benchmark datasets (description can be found in Appendix I), employing the preprocessing method proposed by Papamakarios et al. (2018). In our experiments, we kernelised two flow architectures, i.e., RealNVP and Glow, that utilise the coupling layer for efficient sampling and inference and making direct comparisons with them. Additionally, we also considered comparisons with GF (Gaussianisation Flows) (Meng et al., 2020), RBIG (Rotation-based Iterative Gaussianisation) (Laparra et al., 2011), GIS (Gaussianied Iterative Slicing/Sliced Iterative Normalising Flows) (Dai & Seljak, 2020b), MAF (Masked Autoregressive Flows) (Papamakarios et al., 2018), and MADE (Masked Autoregressive Distribution Estimation) (Germain et al., 2015), FFJORD (Grathwohl et al., 2018), architectures that do not use coupling layers. These methods are not directly comparable to the coupling-layer-based methods, as they have significantly higher computational costs. In particular, Gaussianisation-based methods, require many more layers (up to 100, in our settings), whilst autoregressive flows are slow to sample from, due to their autoregressive nature. In contrast to Gaussianisation-based methods, our kernelisation of coupling layers does not increase the computational complexity and the training time under a fixed number of epochs is similar to neural-net-based coupling layers. Run-time comparisons under a fixed number of epochs are provided in Table 9 in Appendix F),.

Table 2 presents the results of our experiments, revealing that Ferumal flow kernelisation consistently achieves better or competitive outcomes across all five datasets. Despite its straightforward coupling layer architecture, our approach surpasses RBIG, GIS, and MADE on all the datasets and achieves competitive performance to the much more expensive MAF, GF, and FFJORD methods, underscoring the efficacy of integrating kernels. Please refer to Table 12 for error bars on coupling and non-coupling experiments.

## 5.3 INITIAL PERFORMANCE

Figure 3 presents the learning curves of the train and test loss for our Ferumal flow kernelisation and the two neural-network counterparts. These findings demonstrate that the Ferumal-flow-based architecture exhibits faster convergence compared to the neural network baselines. This may be due to the parameter efficiency provided by our kernelisation or due to the stronger inductive biases provided by kernel machines. Throughout our experiments, we maintained default settings and ensured consistent batch sizes across all models.

Table 2: Log-likelihood measured in nats. Larger values are better. Methods prepended with FF are our kernelised versions and the results with * are taken from existing literature.

| | | Datasets | | | | |
| --- | --- | --- | --- | --- | --- | --- |
| | Method | Power | Gas | Hepmass | Miniboone | BSDS300 |
| Coupling | RealNVP | 0.17 | 8.33 | -18.7 | -13.55 | 153.28 |
| | FF-RealNVP (ours) | 0.24 | 9.55 | -18.20 | -11.19 | 154.30 |
| | Glow | 0.17 | 8.15 | -18.92 | -11.35 | 155.07 |
| | FF-Glow (ours) | 0.35 | 10.75 | -17.11 | -10.76 | 154.71 |
| Non-coupling | MADE* | -3.08 | 3.56 | -20.98 | -15.59 | 148.85 |
| | MAF | 0.24 | 10.08 | -17.70 | -11.75 | 155.69 |
| | FFJORD | 0.46 | 8.59 | -14.92 | -10.43 | 157.40 |
| Gaussianisation methods | GF | 0.57 | 10.08 | -17.59 | -10.32 | 152.82 |
| | GIS* | 0.32 | 10.30 | -19.00 | -14.26 | 157.31 |
| | RBIG | -1.02 | -0.05 | -24.59 | -25.41 | 115.96 |

## 5.4 LOW-DATA REGIME

In certain applications, such as medical settings, data availability is often limited. Neural-network-based flows typically suffer from over-parameterisation, leading to challenges in generalisation within low-data regimes. To assess the generalisation capability of our model under such conditions, we trained our model using only 500 examples and evaluated its performance on the same benchmark datasets. To address the challenges of limited data, we opted to tie the learned auxiliary variables across layers in this setting. This approach helped mitigate parameter complexity whilst maintaining the benefits of utilising auxiliary points.

As highlighted by Meng et al. (2020), Glow and RealNVP struggled to generalise in low-data regimes, evidenced by increasing validation and test losses whilst the training losses decreased. To provide a stronger benchmark, we included the FFJORD model (Grathwohl et al., 2018). FFJORD is a continuous normalising flow method with a full-form Jacobian and exhibits superior performance to Glow or RealNVP in density estimation and generation tasks. For our model, we used a ker-

Table 3: Results on a small subset of 500 examples. LL in nats, higher the better

| Dataset | Ours (#params) | FFJORD (#params) |
| --- | --- | --- |
| Miniboone | **-27.75 (58K)** | -39.92 (821K) |
| Hepmass | **-27.90 (41K)** | -28.17 (197K) |
| Gas | **0.22 (11K)** | -7.50 (279K) |
| Power | **-2.91 (8K)** | -11.33 (43K) |
| BSDS300 | **121.22 (85K)** | 100.32 (3,127K) |

nelised version of RealNVP which is notably weaker than the Glow version. This also proves that kernelisation can make flow-based models more data-efficient.

Table 3 presents the results, demonstrating that our method achieves superior generalisation. This may be attributed to the significantly lower number of parameters required compared to the continuous FFJORD method.

## 5.5 PARAMETER EFFICIENCY

Table 4 shows the parameter counts of Ferumal flows against the baseline methods. Kernelising the models results in a parameter reduction of up to 93%. The parameter efficiency consequently results in less data requirement, better generalisation, and faster convergence. This reduction can be further improved by implementing strategies such as sharing auxiliary variables between layers or potentially with low-rank approximations, particularly in scenarios where data is limited and concerns about overfitting arise (see Appendix C for additional details).

Table 4: Number of parameters. Methods prepended with FF are our kernelised versions with % reduction in brackets

| | Architectures | | | |
| Dataset | RealNVP | FF-RealNVP (ours) | Glow | FF-Glow (ours) |
| --- | --- | --- | --- | --- |
| Miniboone | 377K | 117K (69%) | 395K | 141K (64%) |
| Hepmass | 288K | 76K (74%) | 293K | 79K (73%) |
| Gas | 236K | 22K (91%) | 237K | 23K (90%) |
| Power | 228K | 16K (93%) | 228K | 20K (91%) |
| BSDS300 | 458K | 171K (63%) | 497K | 279K (44%) |

# 6 DISCUSSION AND LIMITATIONS

We have introduced Ferumal flows, a novel approach to integrate kernels into flow-based generative models. Our study highlighted that Ferumal flows exhibit faster convergence rates, thanks to the inductive biases imparted by data-dependent initialisation and parameter efficiency. Moreover, we have demonstrated that kernels can significantly reduce the parameter count without compromising the expressive power of the density estimators. Especially in the low-data regime, our method shows superior generalisation capabilities, while Glow and RealNVP fail entirely, and FFJORD lags significantly in performance. We also demonstrate the application of our method in hybrid modelling. (Please refer to Appendix H)

In contrast to neural-network-based flows, kernelised flows require a different hyperparameter selection. In classical kernel machines, the choice of kernel usually implies a type of inductive bias (e.g., for specific data types (Vishwanathan et al., 2010)). Consequently, in this work, we mostly focus on Squared Exponential kernels and Matern kernels, but incorporating kernels with strong inductive biases may be a promising avenue for future research. In particular, parameter sharing for highly structured modalities such as images is another potential direction for future research.

The present work introduces kernels only for some affine coupling layer architectures such as RealNVP and Glow. However, the concepts also directly apply to other coupling-layer-type networks, such as neural spline flows (Durkan et al., 2019), ButterflyFlows (Meng et al., 2022), or invertible attention (Sukthanker et al., 2022) for greater expressiveness and parameter efficiency. Ferumal flow kernelisation can also be directly enhanced with other building blocks such as MixLogCDF-coupling layers (Ho et al., 2019).

Whilst our method can be applied to coupling-type flow-based architectures, it poses challenges when it comes to ResFlow-like architectures Behrmann et al. (2019); Chen & Gopinath (2000), which require explicit control of Lipschitz properties of the residual blocks. As a result, extending our approach to ResFlow-like architectures is left as a direction for future research.

One major drawback of existing normalising flow algorithms is their dependence on an abundance of training data. The introduction of kernels into these models may allow the application of flows in low-data settings. Additionally, in the era of increasingly large and complex models, energy consumption has become a significant concern. Faster convergence can contribute to energy savings. Notably, our models, owing to their faster convergence and few hyperparameters needed fewer training runs than the neural-network counterparts. We anticipate that future research will continue to explore efficient methodologies and strive for reduced energy and data demands.

ACKNOWLEDGMENTS

We extend our gratitude to Noel Danz for his valuable discussions on coding. We would also like to express our appreciation to Arkadiusz Kwasigroch and Alexander Rakowski for their initial feedback on the draft. This research was funded by the HPI research school on Data Science and by the European Commission in the Horizon 2020 project INTERVENE (Grant agreement ID: 101016775)

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

## A    ADDITIONAL EXPERIMENTAL DETAILS FOR OUR METHOD

We employed the Adam optimiser exclusively for all our experiments. The other hyperparameters are chosen using a random grid search, i.e. $lr \in [0.01, 0.005, 0.001]$, $\beta_1, \beta_2 \in [0.85, 0.9, 0.95, 0.99]$, kernel $\in [matern32, matern52, rbf]$. We used Cosine decay for all experiments with the minimum learning rate equalling zero. In initial experiments, we found that 150 auxiliary points performed satisfactorily and we persisted with it for the majority of datasets and tried 200 for datasets with high-dimensionality. For comprehensive information, please refer to Table 5.

For the experiments presented in Table 1, we employed Glow-based architectures for both approaches. We trained these datasets for 1000 training steps, with the data synthesised at every training step (as done in existing implementations). For experiments in Table 1, Table 3, we exclusively use the RBF/Squared-Exponential kernel and randomly sampled the kernel length scale from a log-uniform distribution, i.e., $\gamma \sim \exp(U)$, where $U \sim \mathcal{U}_{[-2,2]}$.

Table 5: Model Architectures and hyperparameters for our method.

| Method | Datasets | | | | |
|---|---|---|---|---|---|
| | Power | Gas | Hepmass | Miniboone | BSDS300 |
| Dimensionality | 6 | 8 | 21 | 43 | 63 |
| Training Points | 1,615,917 | 852,174 | 315,123 | 29,556 | 1,000,000 |
| layers | 14 | 12 | 15 | 12 | 12 |
| kernel | matern52 | matern32 | matern32 | matern52 | rbf |
| auxiliary points | 150 | 150 | 150 | 150 | 200 |
| learning rate(lr) | 0.005 | 0.005 | 0.01 | 0.005 | 0.005 |
| $\beta_1$ | 0.95 | 0.9 | 0.99 | 0.99 | 0.95 |
| $\beta_2$ | 0.9 | 0.99 | 0.99 | 0.90 | 0.85 |
| lr schedular | cosine | cosine | cosine | cosine | cosine |
| min lr | 0 | 0.95 | 0.95 | 0.95 | 0 |
| epochs | 200 | 200 | 500 | 600 | 400 |
| batchsize | 2000 | 2000 | 1024 | 2000 | 1024 |

## B    MODELLING DISCONTINUOUS DENSITIES

We consider two toy datasets with discontinuous densities, CheckerBoard and Diamond. We used 4 flow steps for all the models and piecewise polynomial kernel and learnt its hyperparameters. Figure 2 showcases the original data samples(Left) alongside the modelled density(Middle) and the samples generated using the flow(Right). The neural-net-based architecture for

Table 6: Results on discontinuous densities. Log Likelihood in nats, higher is better

| Dataset | Ours (#params) | NN-based (#params) |
|---|---|---|
| Checkerboard | **-3.61 (1.8K)** | -3.68 (18.4K) |
| Diamond | **-3.24  (1.8K)** | -3.39 (18.4K) |

the checkerboard dataset in the first row shows blurry boundaries and ill-defined corners. The kernelised counterpart in the second row has better-defined boundaries in some cases. However, this effect is even more pronounced in the diamond dataset with the kernelised counterpart in the fourth row modelling the discontinuity way better. Table 2 shows the corresponding log-likelihood in nats, quantitatively showing the enhancement from our kernelisation. The results demonstrate that Ferumal flow kernelisation can outperform its neural net counterpart on these toy datasets. All the toy datasets were trained with a batch size of 512 and for 100K iterations. We found that the piecewise polynomial kernel was better suited for discontinuous densities than Matern kernels. This provides another evidence of better performance due to inductive biases of a kernel.

1. Neural-Net-based flow for the Checkerboard dataset

2. Our kernelised flow for the Checkerboard dataset

3. Neural-Net-based flow for the Diamond dataset

4. Our Kernelised flow for the Diamond dataset

Figure 2: Discontinuous distributions. Shown are training data (left column), flow density (center column), and and histogram of flow samples (right column)

Table 7: Results on a small subset of 500 examples. Log-Likelihood in nats, Higher the better

| Dataset | shared auxiliary (#params) | low rank (#params) | no auxiliary (#params) |
|---|---|---|---|
| Miniboone | -27.75 (58K) | -28.53 (19K) | -41.83 (345K) |
| Hepmass | -27.90 (41K) | -27.91 (14K) | -29.01 (126K) |
| Gas | +0.22 (11K) | -1.85 (6K) | -10.19 (32K) |
| Power | -2.91 (8K) | -3.13 (6K) | -9.26 (36K) |
| BSDS300 | +121.22 (85K) | +111.91 (17K) | +109.265 (505K) |

## C  LOW-RANK APPROXIMATIONS

For datasets characterised by high dimensionality and complex structures, relying solely on auxiliary points for the weight matrix ($A \in \mathbb{R}^{2(D-d) \times N}$) is inefficient. When half of the dimensions are transformed (as is the case in any coupling layer), this matrix becomes excessively large.

To preserve the desirable quality of generalisation in our models, we propose an alternative approach to obtaining the weight matrix for a kernelised layer. We suggest using the product of two smaller matrices (with fewer parameters) instead. For a weight matrix $A \in \mathbb{R}^{p \times N}$, responsible for producing $p$ affine parameters using $N$ auxiliary points, we can learn two smaller matrices: $A^1 \in \mathbb{R}^{c \times N}$ and $A^2 \in \mathbb{R}^{c \times p}$ where $c < p$. By employing the outer product $\hat{A} = A^{2\top} A^1 \approx A$ as a proxy for a full-weight matrix, we can effectively reduce the number of parameters.

In Table 7, we present the effectiveness of employing a low-rank approximation on the identical subset of 500 samples, as depicted in Table 3 in the main manuscript. This technique ensures a minimal number of parameters while achieving satisfactory generalisation. During the experimental setup, we endeavoured to utilise the lowest feasible value of $c$ (chosen via a hyperparameter grid of $\{4, 8, 12\}$) that would still yield reliable generalisation. Notably, our approach achieves good results while providing superior control over the parameters, in contrast to the shared auxiliary variable method.

**Learning without auxiliary points**   We also present a comparison of another variation of our method, i.e., learning without the use of auxiliary variables. Notably, the sharing of auxiliary variables yields the best results, followed by the utilisation of low-rank matrices in conjunction with auxiliary variables. Whilst the results obtained with low matrices are not significantly different from those obtained with shared auxiliary variables, they offer a further reduction in parameters. As seen in Table 7, not using auxiliary variables causes a high number of parameters and the model overfits easily causing comparatively poor results (notably fewer parameters than FFJORD, depicted in Table 3 in the main manuscript, while achieving somewhat similar performance).

## D  DENSITY ESTIMATION ON REAL WORLD MEDICAL DATASET

Following our experiments within the low-data regime, as discussed in Section 5.4, we extend our analysis to a real-world medical dataset—the UK Biobank (Bycroft et al., 2018; Kirchler et al., 2022). This dataset encompasses phenotype and genotype information for a substantial cross-section of the UK population, encompassing a total of 30 biomarkers. Notably, only 3,240 individuals within the dataset possess complete information on all biomarkers.

In line with our experiments in Section 5.4, we conducted a comparative analysis between our kernelised-RealNVP and FFJORD. The density estimation results presented in Table 8 exhibit better performance whilst training significantly faster, thereby reinforcing the findings in Table 3.

Table 8: Results on the UKBiobank's biomarker data. Log Likelihood in nats, higher is better

| Method | Ours | params | train time |
|---|---|---|---|
| **Ours** | **-29.11** | **41K** | **21 min** |
| FFJORD | -31.01 | 1.1M | 13.1 hrs |

# E    DISCUSSION ON OUR USE OF KERNELS

**Effect of kernel length scale**   As with any kernel machine, the length scale serves as a highly sensitive hyperparameter in our method. During our investigation, we discovered that in identical settings, distinct kernel length scales produce varying outcomes. Certain scales have a tendency to overfit easily on the training set (e.g., -3.76 nats training likelihood on Miniboone), while some tend to underfit (e.g.,-40.23 nats training likelihood on Miniboone). This diverges from neural-net-based flows, where overfitting necessitates additional layers or nodes within each layer (bearing in mind that this results in an increase in parameters unlike our method). Such findings vividly demonstrate the high expressiveness of kernelisation in flow-based models.

**Composite kernels**   In our experiments, we mostly employed the Squared Exponential Kernel/RBF, Matern kernels. Nevertheless, it is feasible to employ a combination of kernels, also known as multiple kernel learning (Sonnenburg et al., 2006). We defer the comprehensive analysis of kernel composition and its application to future endeavours.

# F    RUNTIME COMPARISONS

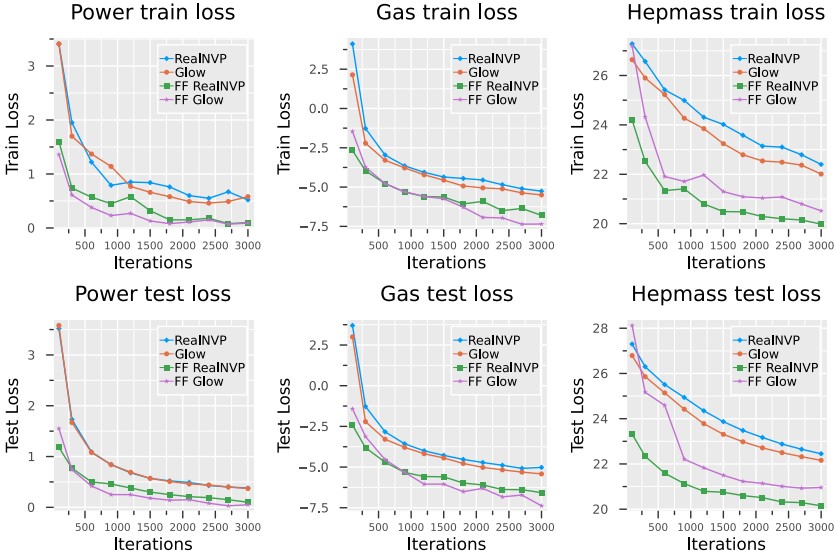

Figure 3: Negative log-likelihood (loss in nats) on training and test sets over the first 3,000 training iterations. Methods prepended with FF are our kernelised versions. All models were further trained until convergence.

Table 9: Training time, hours and minutes

| Dataset | Glow | Ours | FFJORD | GF |
|---------|------|------|--------|-----|
| Miniboone | 63m | 51m | 5h 21m | 4h 37 m |
| Hepmass | 8h 23m | 8h 49m | > 1 day | > 1 day |
| Gas | 8h 45m | 9h 07m | > 1 day | > 1 day |
| Power | 8h 13m | 8h 50m | > 1 day | > 1 day |
| BSDS300 | 10h 57m | 10 49m | > 1 day | > 1 day |

We picked the best models in each category from Table 2 and compared them with our kernelised Glow model. For fair comparisons, we ran the models for the same number of epochs and used the same number of flow steps for the neural-net counterpart, Glow model. It is worth noting that our model has faster convergence up to 3 times than the Glow model. Table 9 shows that despite using kernels, we have comparable run times with the neural-network counterpart. We perform significantly

better than continuous time normalising flow, FFJORD, and iterative-gaussianisation-based kernel method, Gaussianisation flow, both take longer than a day on bigger datasets.

## G    IMPROVING MIXERFLOW'S IMAGE GENERATION

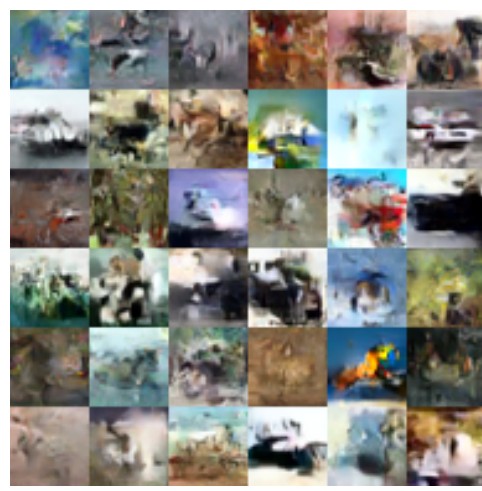

(a)  NN-based MixerFlow

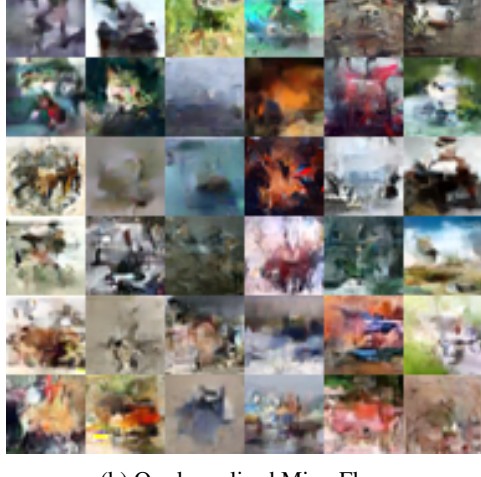

(b) Our kernelised MixerFlow

Figure 4: Sampled images from MixerFlow

MixerFlow (English et al., 2023) is a flow architecture for image modelling. Unlike Glow-based architectures that relies on convolutional neural networks for image generation, MixerFlow offers a flexible way to integrate any flow method, making it suitable for image generation with our kernelised coupling layers. In this section, we showcase the application of our kernelisation to the MixerFlow model on the CIFAR-10 dataset (Krizhevsky, 2009b). We employed small models(30 layers) and trained on a single gpu. For fair comparison, we kept the same model architecture with the sole change being the replacement of the neural-network-based coupling layers with our kernelised coupling layer. Our results as shown in Table 10 demonstrate kernelisation can yield better result with faster convergence attributed to reduction of parameters. The generated samples can be seen in Figure 4a and  4b.

Table 10: Results on the CIFAR-10 dataset. Log Likelihood in nats, higher is better

| Method | Ours | params | convergence step |
|---|---|---|---|
| **Ours(kernelised)** | **-7644.39** | **4.85M** | **103K** |
| NN-based MixerFlow | -7665.65 | 11.43M | 195K |

## H    IMPROVING VAE'S ELBO

We also try our kernelised flows in hybrid settings, demonstrating that we can integrate them with neural-net-based architectures. We apply our kernelised model, FF-GLow, to the variational autoencoder (Kingma & Welling, 2022) in the form of flexible prior and approximate posterior distributions. We apply the methods to Kuzushiji-MNIST, which is a variant of MNIST containing Japanese script. We investigate the capacity of our kernelisation to improve over the baseline of standard-normal prior and diagonal-normal approximate posterior, and its neural network counterpart, Glow. We use 6 flow steps for each flow-based model and the latent hidden dimension equals 16. The quantitative results are shown in Table 11 and generated image samples in Figure 7

Both models (Glow and ours FF-Glow) improve significantly over the standard baseline. However, there is no considerable quantitative gain by using the kernelised version. We believe that this might

be due to the Glow model being sufficient to model the latent space on the dataset and having a little margin for kernelisation to shine. However, our kernelisation still helps in making the model parameter efficient with only a small increase in parameter complexity compared to the baseline.

Table 11: VAE test-set results (in nats) for the evidence lower bound (ELBO) on the Kuzushiji-Mnist dataset. Error bars correspond to two standard deviations.

|  | ELBO | $\log p(x)$ | Params |
|---|---|---|---|
| Baseline | -195.61±1.25 | -182.33± 1.30 | 1.18M |
| Glow | -189.99±1.35 | -178.89±1.25 | 2.05M |
| FF-Glow (ours) | -189.48±1.36 | -178.54±1.25 | 1.23M |

# I  DETAILS OF THE DATASETS

In the following paragraphs, a brief description of the five datasets used in Table 2 (POWER, GAS, HEPMASS, MINIBOONE, BSDS300) and their preprocessing methods is provided.

Table 12: Log-likelihood measured in nats. larger values are better. Methods prepended with FF are our kernelised versions and the results with * are taken from existing literature. Error bars correspond to 2 standard deviations.

|  | Method | Datasets | | | | |
|---|---|---|---|---|---|---|
|  |  | Power | Gas | Hepmass | Miniboone | BSDS300 |
| Coupling | RealNVP | 0.17± 0.01 | 8.33± 0.14 | -18.7± 0.02 | -13.55± 0.49 | 153.28± 1.78 |
|  | FF-RealNVP (ours) | 0.24± 0.01 | 9.55±0.03 | -18.20±0.04 | 1-1.19± 0.35 | 154.30± 2.11 |
|  | Glow | 0.17± 0.01 | 8.15± 0.40 | -18.92±0.08 | -11.35±0.07 | 155.07 ± 1.03 |
|  | FF-Glow (ours) | 0.35± 0.01 | 10.75± 0.02 | -17.11±0.02 | -10.76±0.44 | 154.71± 0.28 |
| Non-coupling | MADE* | -3.08 ± 0.03 | 3.56± 0.04 | -20.98± 0.02 | -15.59± 0.50 | 148.85± 0.28 |
|  | MAF | 0.24± 0.01 | 10.08± 0.02 | -17.70± 0.02 | -11.75± 0.44 | 155.69 ± 0.28 |
|  | FFJORD | 0.46 ± 0.01 | 8.59± 0.12 | -14.92 ± 0.08 | -10.43± 0.04 | 157.40± 0.19 |

**POWER:**  The POWER dataset comprises measurements of electric power consumption in a household spanning 47 months. Although it is essentially a time series, each example is treated as an independent and identically distributed (i.i.d.) sample from the marginal distribution. The time component was converted into an integer representing the number of minutes in a day, followed by the addition of uniform random noise. The date information is omitted and the global reactive power parameter, as it had numerous zero values that could potentially introduce large spikes in the learned distribution. Uniform random noise was also added to each feature within the interval $[0, \epsilon_i]$, where $\epsilon_i$ is chosen to ensure that there are likely no duplicate values for the i-th feature while maintaining the integrity of the data values.

**GAS:**  The GAS dataset records readings from an array of 16 chemical sensors exposed to gas mixtures over a 12-hour period. Like the POWER dataset, it is essentially a time series but was treated as if each example followed an i.i.d. distribution. The data selected represents a mixture of ethylene and carbon monoxide. After removing strongly correlated attributes, the dataset's dimensionality was reduced to 8.

**HEPMASS:**  The HEPMASS dataset characterizes particle collisions in high-energy physics. Half of the data correspond to particle-producing collisions (positive), while the remaining data originate from a background source (negative). In this analysis, we utilized the positive examples from the "1000" dataset, where the particle mass is set to 1000. To prevent density spikes and misleading results, five features with frequently recurring values were excluded.

**MINIBOONE:**  The MINIBOONE dataset is derived from the MiniBooNE experiment at Fermilab. Similar to HEPMASS, it comprises positive examples (electron neutrinos) and negative examples

(muon neutrinos). In this case, only the positive examples were employed. Some evident outliers (11) with values consistently set to -1000 across all columns were identified and removed. Additionally, seven other features underwent preprocessing to enhance data quality.

**BSDS300:** The dataset was created by selecting random 8x8 monochrome patches from the BSDS300 dataset, which contains natural images. Initially, uniform noise was introduced to dequantize the pixel values, after which they were rescaled to fall within the range [0, 1]. Furthermore, the average pixel value was subtracted from each patch, and the pixel located in the bottom-right corner was omitted.

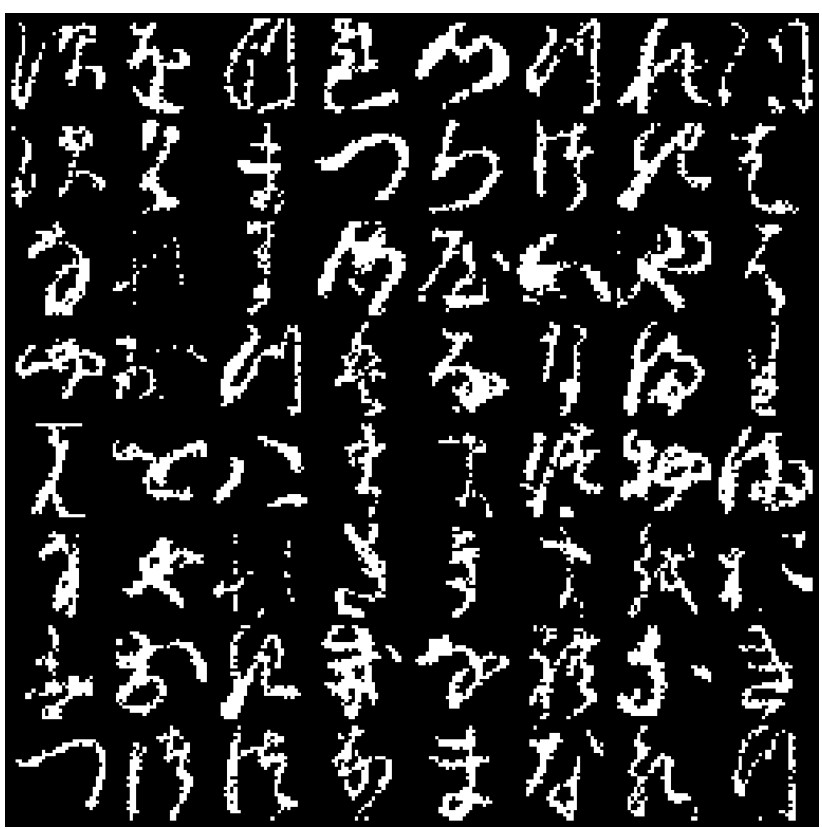

Figure 5: Original samples after binarisation

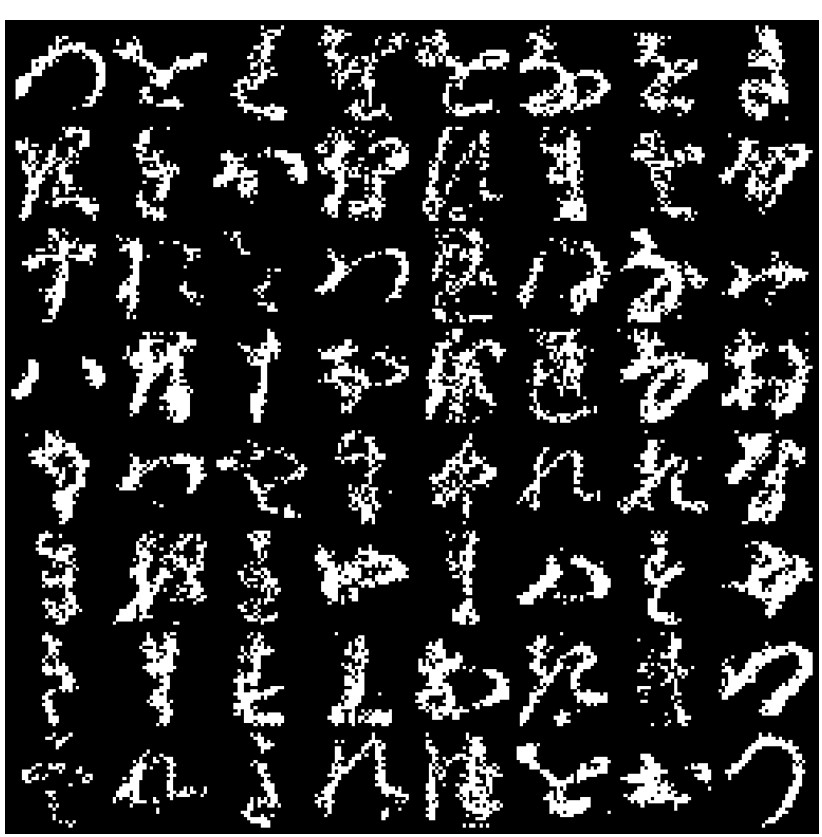

Figure 6: VAE samples from neural-net-based Glow

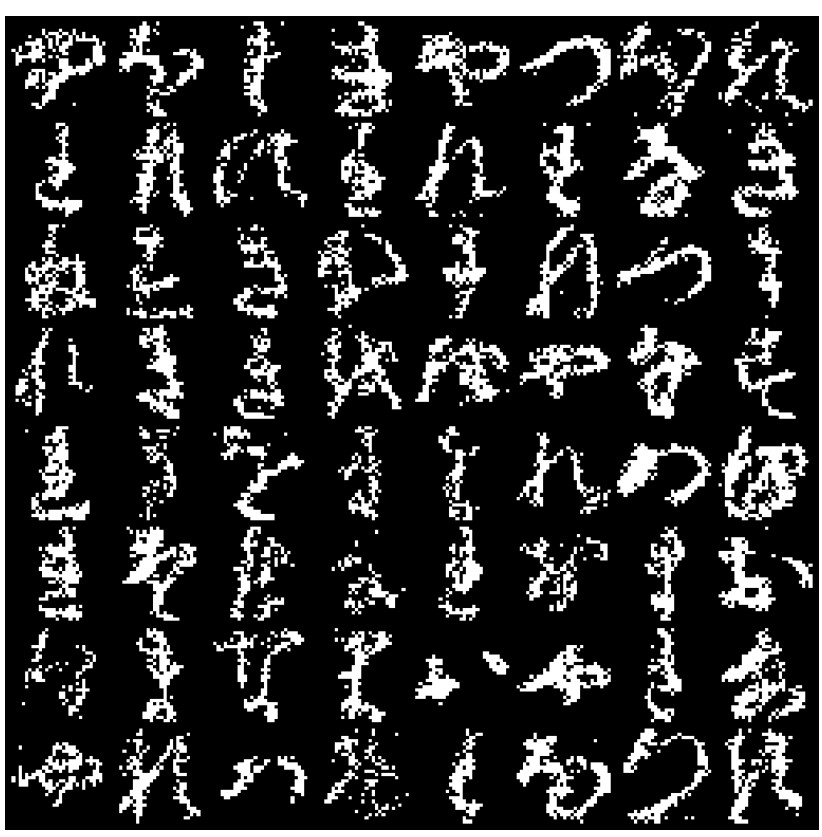

Figure 7: VAE samples from our kernelised flow, FF-Glow

