# OpenReview forum: "Kernelised Normalising Flows"
_ICLR.cc/2024/Conference — ICLR 2024 poster_

### Official Review · Reviewer_UUvL · 2023-10-29

**Soundness:** 3 good
**Presentation:** 3 good
**Contribution:** 3 good
**Rating:** 8
**Confidence:** 4

**Summary:**

Update after rebuttal:
I thank the authors for their response. The clarifications around the representer theorem and tabular / non image data were helpful. I encourage the authors to include the promised changes in their updated manuscript. I read the other reviews and rebuttal and do not see the negative points raised as a significant reason to reject this paper. I have raised my score to an 8.


Normalising flows are probabilistic machine learning models that are capable of jointly solving the task of density estimation and generative modelling (sampling). Unfortunately, since they are parameterised in terms of a pushforward and their log-likelihood is determined using a change-of-variables formula involving a Jacobian, the mappings involved must be invertible, and are often even more constrained (e.g. through coupling layers). Invertibility imposes a big constraint on the model, meaning that the only lever one may pull to obtain expressiveness is often the depth and overparameterisation of the neural network. This results in data-hungriness, and makes them highly unsuitable for modelling tabular and low-dimensional data.

The authors introduce a kernelised version of normalising flows, which are suitable for modelling low-dimensional and tabular data. Efficacy is demonstrated on a set of benchmark datasets.

**Strengths:**

- The paper ideas presented in this paper are conceptually simple and do not require a strong leap-of-faith on the reader's end.
- The mathematical idea (a representer theorem) appears to be mostly sound.
- The experiments on tabular / low dimensional data are convincing and demonstrate the appeal of the method. A suitable benchmark is considered.
- The paper is well-written. Text, equations, tables and figures are appropriately laid out and given the right amount of real-estate.

**Weaknesses:**

- I believe the following statement, after the proof of Proposition 3.1, is misleading (or possibly even incorrect): "Note that, in contrast to the classical representer theorem, since the objective doesn’t contain a strictly convex regularisation term that penalises the model complexity, the solution is not necessarily unique". I believe the authors might be referring to Theorem 1 of Scholkopf et al. 2001. In this classical representer theorem, the objective need not be convex, nor does the regularisation term need to be complex, nor does the solution need to be unique, nor does a convex regularisation term even necessarily enforce that the solution is unique (or, more weakly, that the objective is convex). The theorem simply says that any minimiser (if it exists), must be a finite linear combination of the kernel evaluations. It is true that in many typical applications of the representer theorem (e.g. kernel ridge regression, L2 penalised kernel logistic regression), the loss is strictly convex and the regularisation is strongly convex so the overall objective is strongly convex. However more generally, the theorem applies to non-convex objectives.
- "This makes kernelised flows especially promising for learning in the low-data regime, as their model complexity naturally scales with dataset size and does not over-parametrise as much as neural networks", however there remains the previously mentioned caveat that the number of layers L is still a hyperparameter and increases model complexity. One can still over-parameterise with increasing L, or am I wrong?

I am willing to update my scores if the authors can respond to these weaknesses, and also help me understand the answers to my questions below.

**Questions:**

Could the authors please clarify the weaknesses mentioned above? Specifically:
- Is the discussion around the representer theorem accurate?
- Can the model be overparameterised by increasing L, and how does this fit into the discussion around suitability of non-overparameterised/data hungry models for tabular / low dimensional data regimes?

Additional questions (neither strengths nor weaknesses)
- I missed why the method is called "Ferumal". Can the authors quickly explain (apologies if this is mentioned somewhere already in the paper)?
- Is there any method of constraining the support of the distribution (e.g. to lie on the simplex or some other set)? Perhaps this is as simple as constraining the support of the base distribution p_Z?
- "Our study highlighted that Ferumal flows exhibit faster convergence rates, thanks to the inductive biases imparted by data-dependent initialisation and parameter efficiency." I missed where the authors talked about data-dependent initialisation. Is this because the kernel matrix is seen as a parameter, which is initialised directly from the data, and then the auxiliary points are learned?
- Future work. Is there a possibility to use deep kernel learning to learn the kernel as well? Would this be possible under MLE / MAP estimation, or is another objective required?
- Future work. Can these be trained in a Bayesian manner? If the ELBO is used as an objective, can the representer theorem still be applied?
- Open question: When is density estimation/sample generation of tabular data of practical concern in a machine learning context? I can see how sample generation is particularly interesting for high dimensional structured data (e.g. images).


Related works:
The authors might like to mention the more recent related literature of kernel methods for nonnegative functions as a way of building kernel-based probability density functions, which are also applicable and demonstrated in low dimensional / tabular data regimes, as well as their related neural-network based models through the NNGP. These also do not require invertibility. For example:
- PSD representations for effective probability models, NeurIPS 2021
- Sampling from arbitrary functions via PSD models, AISTATS 2022
- Squared Neural Families: A New Class of Tractable Density Models, NeurIPS 2023

---

> ### Author Response · Authors · 2023-11-18
>
> **"Is the discussion around the representer theorem accurate?"**
>
> Thank you for pointing out this important distinction; as you correctly recognised, the sentence in question was incorrect in its original form. We reformulated the sentence as follows:
> "In contrast to the classical representer theorem, the objective doesn't contain a regulariser of the model's norm, which would ensure that any solution can necessarily be represented as a linear combination of kernel evaluations.
> However, if a solution exists, Proposition 3.1 ensures that there also exists a solution that can be expressed as a linear combination of kernel evaluations."
> The remaining discussion of Proposition 3.1, including the kernelisation of the objective, is not affected by the change.
>
> **""This makes kernelised flows especially promising for learning in the low-data regime, as their model complexity naturally scales with dataset size and does not over-parametrise as much as neural networks", however there remains the previously mentioned caveat that the number of layers L is still a hyperparameter and increases model complexity. One can still over-parameterise with increasing L, or am I wrong?"**
>
> You are certainly correct in that an excessive number of layers can still lead to overparametrisation in our method. As the context of the sentence you referenced indicates, this only pertains to the "per-layer complexity" of kernelised flows. We added a qualification to the paragraph. Additonally, we would like to point that overparameterisation in the context of neural-net-based coupling layers is primarily attributed to deep neural networks within layers(per-layer complexity,k). In contrast use of shallow networks in coupling layers often lack the expressive power needed for effective scaling and translation parameters.
>
> **"Is there any method of constraining the support of the distribution (e.g. to lie on the simplex or some other set)? Perhaps this is as simple as constraining the support of the base distribution p_Z?"**
>
>  The relation between base and target distribution is complex. Due to invertibility, the topology of the support between base and target distribution remains unchanged. Recent work by Stimper et al (2022) proposed a resampling scheme of the base distribution that allows for better modelling of distributions with disconnected components. These methods should directly translate to our kernelised flows, as they are architecture-independent. However, we are not aware of any work to directly constrain the target support; restricting the base distribution will not suffice, as the flow network can simply re-scale the features.
>
> **"Our study highlighted that Ferumal flows exhibit faster convergence rates, thanks to the inductive biases imparted by data-dependent initialisation and parameter efficiency."
> I missed where the authors talked about data-dependent initialisation. Is this because the kernel matrix is seen as a parameter, which is initialised directly from the data, and then the auxiliary points are learned?"**
>
> Yes, since the auxiliary points are initialised directly from the data and then learned, we refer to it as data-dependent initialisation.
>
> **Future work. Is there a possibility to use deep kernel learning to learn the kernel as well? Would this be possible under MLE / MAP estimation, or is another objective required?**
>
> Our method in its current form allows for any kind of fixed kernels, including deep kernels. Prop 3.1 only applies to fixed kernels, so simply optimising over the deep kernel parameters would at least invalidate the interpretation of the method as a multi-layered kernel machine and might lead to instable training and overparametrisation. We expect more classical multiple kernel learning with learnable weights to be much easier to integrate into the objective.
>
> **"I missed why the method is called "Ferumal". Can the authors quickly explain (apologies if this is mentioned somewhere already in the paper)?"**
>
> It was named in the loving memory of the grandfather of one of the authors.
>
> **"Related works: The authors might like to mention the more recent related literature of kernel methods for nonnegative functions as a way of building kernel-based probability density functions, which are also applicable and demonstrated in low dimensional / tabular data regimes, as well as their related neural-network based models through the NNGP. These also do not require invertibility.**
>
>  Thanks for the suggested references, we will add them into the updated manuscript.

---

> ### Author Response · Authors · 2023-11-18
>
> **"Future work. Can these be trained in a Bayesian manner? If the ELBO is used as an objective, can the representer theorem still be applied?"**
>
>  Thank you for this interesting question. As of now, we don't see that normalising flows in general are well-established for Bayesian density estimation. Any such approach would require setting appropriate density priors. Especially in our kernel setting, an interesting starting point for this could be the work on Gaussian Processes, and especially on GP density samplers (Adams et al, (2008)), as GPs are natural extensions of classical kernel machines to Bayesian settings. For conditional density estimation, Trippe & Turner introduced conditional Bayesian normalising flows which are trained via stochastic variational inference similar to other Bayesian neural networks; we would expect similar approaches to work for our kernelised normalising flows. On a related note, in the appendix, in section E, we already show that our kernelised flows can be used as the latent distribution for VAEs.
>
> **"When is density estimation/sample generation of tabular data of practical concern in a machine learning context? I can see how sample generation is particularly interesting for high dimensional structured data (e.g. images)."**
>
> This question pertains to the use of normalising flows on non-imaging data in general. We believe there are a large number of interesting applications for normalising flows. For example, Hansen et al (2022) use normalising flows to model the distribution for feature selection with Model-X knockoffs; Kirchler et al (2023) use the normalisation property of flows to enable multi-variate hypothesis testing on biomarker data in genome-wide association studies. Flows can also be used to model non-parametric financial stock market data for different trading strategies (Kirchler et al (2023) and Huang et al (2023)). Additionally, in the medical domain, generating synthetic human health data is crucial for sharing tabular health data without compromising privacy.
>
> **References**
>
>
> Stimper, Vincent, Bernhard Schölkopf, and José Miguel Hernández-Lobato. "Resampling base distributions of normalizing flows." _International Conference on Artificial Intelligence and Statistics_. PMLR, 2022.
>
> Hansen, Derek, Brian Manzo, and Jeffrey Regier. "Normalizing Flows for Knockoff-free Controlled Feature Selection." _Advances in Neural Information Processing Systems_ 35 (2022): 16125-16137.
>
> Kirchler, Matthias, Christoph Lippert, and Marius Kloft. "Training normalizing flows from dependent data." _International Conference on Machine Learning_. PMLR, 2023.
>
> Huang, Huifang, et al. "Model-based reinforcement learning with non-Gaussian environment dynamics and its application to portfolio optimization." _Chaos: An Interdisciplinary Journal of Nonlinear Science_ 33.8 (2023).
>
> Dutordoir, Vincent, et al. "Gaussian process conditional density estimation." _Advances in neural information processing systems_ 31 (2018).
>
> Trippe, Brian L., and Richard E. Turner. "Conditional density estimation with bayesian normalising flows." _arXiv preprint arXiv:1802.04908_ (2018).

---

> ### Comment · Area_Chair_9TfK · 2023-11-20
> **Respond to authors' rebuttal**
>
> Please, confirm that you have read the author's response and the other reviewers' comments and indicate if you are willing to revise your rating.

---

> ### Comment · Reviewer_UUvL · 2023-11-20
>
> I thank the authors for their response. The clarifications around the representer theorem and tabular / non image data were helpful. I encourage the authors to include the promised changes in their updated manuscript. I read the other reviews and rebuttal and do not see the negative points raised as a significant reason to reject this paper. I have raised my score to an 8.

---

> > ### Author Response · Authors · 2023-11-20
> >
> > We are grateful for the score update and will incorporate the changes in the updated manuscript. Thanks.

---

### Official Review · Reviewer_wvm3 · 2023-10-31

**Soundness:** 3 good
**Presentation:** 3 good
**Contribution:** 2 fair
**Rating:** 6
**Confidence:** 4

**Summary:**

This work focuses on developing normalizing flow-based approaches for cases where we have limited number of samples from the data distribution. Instead of neural network based transformations in coupling layers, the proposed method relies on kernel based transformations to reduce the number of samples required for convergence. The proposed approach slows promising results on 2D synthetic data and five real-world tabular benchmark datasets.

**Strengths:**

The proposed approach is novel and interesting. The paper correctly claims that normalizing flows which employ neural network based coupling layers are data and parameter hungry. The proposed kernel based approach in contrast is data and parameter efficient.

·         The results in Table 3 show that the proposed approach shines in the low data regime. It clearly outperforms FFJORD and obtains impressive results even when only 500 data samples are available.

·         The paper is well written and theoretically well founded. Proposition 3.1 is especially interesting as it states that the model  complexity of the proposed approach scales with naturally with the dataset size. This shows that the proposed approach is not over-parametrized like neural network based approaches and thus should be more suitable for the low-data regime.

**Weaknesses:**

·         Methods like FFJORD (\cf Figure 2 in FFJORD) report better results compared to the proposed approach (as shown in Table 1). The performance advantage of FFJORD is even more apparent in case of the challenging discontinuous checkerboard dataset in Figure 3 (supplementary). It is not clear if the proposed model has the modelling capacity to capture complex distributions.

·         The proposed method is outperformed significantly by FFJORD, although FFJORD uses more parameters as reported in Table 4. Can the performance of the proposed method be improved in Table 4 by increasing the number of parameters?

·         For the experiments on the 2D toy datasets in Table 1, it is not clear which NN based approach is employed. Furthermore, the number of data samples used for training for all models in Table 1 should also be reported.

·         The method is motivated by alluding to medical settings in Section 1 and 5.4, where data availability is often limited. However, the method is never applied to any medical data.

·         Finally, it not clear if the proposed model can be applied to complex data distributions such as images. The qualitative examples of samples generated by the proposed model trained on the Kuzushiji-MNIST dataset as shown in Figure 4-6 are not promising.

**Questions:**

The paper should include a more detailed comparison with FFJORD  especially in case of 2D synthetic datasets such as the discontinuous checkerboard dataset.

·         The paper should discuss in more detail if the performance of the proposed approach can be further boosted by scaling the number of parameters.

·         The paper should discuss in more detail if the proposed approach is applicable to complex datasets such as images.

---

> ### Author Response · Authors · 2023-11-18
>
> **"Methods like FFJORD (\cf Figure 2 in FFJORD) report better results compared to the proposed approach (as shown in Table 1). The performance advantage of FFJORD is even more apparent in case of the challenging discontinuous checkerboard dataset in Figure 3 (supplementary). It is not clear if the proposed model has the modelling capacity to capture complex distributions."**
>
>  In the toy experiments(Figure 1), our focus was on illustrating the effectiveness of kernelisation under the same architecture.  With toy examples in the discontinued densities (figure 3) our objective was to showcase how composite kernels within the same architecture could introduce inductive biases. The performance could be further enhanced by exploring composition of kernels or with more data. Our comparisons in the toy datasets used the same architecture (with and without our kernelisation).
>
>  We acknowledge that FFJORD in some cases can yield better performance, but it should be noted that FFJORD is a continuous time normalising and the performance gain comes with very high training time as reported in Table 10. Throughout our comparisons(especially for the toy datasets), we primarily contrasted our approach with unkernelised versions of the same architectures, unless non-feasible, as seen in the low data regime experiments where overparameterisation led to overfitting, leading us to use FFJORD(having better performance than Glow or RealNVP as shown in Table 2) for meaningful comparison.
>
> **"The proposed method is outperformed significantly by FFJORD, although FFJORD uses more parameters as reported in Table 4. Can the performance of the proposed method be improved in Table 4 by increasing the number of parameters?"**
>
>  We would like to highlight that whilst FFJORD significantly outperforms on large-datasets (except Gas and Miniboone dataset) , it under performs when the number of datapoints is really low as shown in Table 4. Additionally, FFJORD takes significantly longer training periods even for small/toy datasets. This efficiency aspect is a crucial consideration, especially in scenarios where computational resources are limited or training time is a critical factor.
>
>  Regarding the question of improving our method's performance by increasing the number of parameters, we posit that the kernelised architectures may derive advantages, especially when the increased parameters result from adding more layers(similar to neural-net counterparts) rather than learning an excess of auxiliary points. However, a more promising direction would be using the composite kernels (or even deep kernel learning). As discussed in the discontinued densities section, we briefly illustrate how simple models, inherently unable to capture complex distributions, can be readily augmented by composite kernels to enhance performance. We believe, this holds substantial promise for further advancements of kernelised architectures.
>
> **"For the experiments on the 2D toy datasets in Table 1, it is not clear which NN based approach is employed. Furthermore, the number of data samples used for training for all models in Table 1 should also be reported."**
>
> In the experiments on the 2D toy datasets presented in Table 1, we employed Glow-based architectures for both approaches. We trained these datasets for 1000 training steps, with the data synthesised at every training step (as done in existing implementations). We will add these details in an updated version of the manuscript.
>
> **"The method is motivated by alluding to medical settings in Section 1 and 5.4, where data availability is often limited. However, the method is never applied to any medical data."**
>
>  We have now incorporated an evaluation of our method's performance on a medical dataset, demonstrating its practical applicability in a real life application. Please refer to Section D of the appendix for the results..
>
>  **"The qualitative examples of samples generated by the proposed model trained on the Kuzushiji-MNIST dataset as shown in Figure 4-6 are not promising."**
>
>  The primary objective of our VAE experiments was to demonstrate improvement over the Evidence Lower Bound (ELBO) in the specified settings. It's important to note that the small latent dimension in our experimental setup impacts the visualisation quality and a bigger latent dimension could potentially improve the quality of generated samples.

---

> ### Author Response · Authors · 2023-11-18
>
> **"The paper should discuss in more detail if the proposed approach is applicable to complex datasets such as images."**
>
>  The primary focus in this work was on making normalising flows work well in the low-data regime, where we believe our method has significant potential and practical applications. Notably, our proposed approach serves as a direct replacement for MLP-based coupling layers, traditionally employed for non-image data. In contrast, Glow-based models use to the convolutional neural networks (CNNs) for image generation. Adapting the current method for convolutional neural networks is beyond the scope of current study and reserved for future works.
>
>  However, we showcase image generation via applicability of our kernelisation to the MixerFlow architecture, a coupling layer architecture for image generation that does not rely on convolutions for weight sharing. Our kernelised method can be directly applied to coupling layer architectures for image generation that do not rely on convolutions for weight sharing. The results of this application are presented in appendix E. It's important to note that for fair comparison, we maintained the same architecture, demonstrating that kernelisation can be successfully applied to image generation tasks with improvements over the neural-net-based architecture. Additionally, we would like to highlight that visually better image generation with flows require larger models and are typically trained on multiple GPUs. Our tabular models are trained on CPUs whereas image models are trained on a single Colab GPUs. Consequently, we have evaluated smaller models for image generation. Training bigger models with fine-grained hyperpaprameter tuning can help generate better visualisation, which is typically the case with flow-based image generation.

---

> > ### Author Response · Authors · 2023-11-22
> >
> > Thank you for your time and effort in reviewing our work. In case, you require further clarification on our answers, we are happy to provide it. If we have addressed your concerns, would you kindly consider revising your score?
> >
> > Thanks!

---

> > > ### Comment · Reviewer_wvm3 · 2023-11-22
> > > **Thanks for the rebuttal**
> > >
> > > Thank you for including the results on the medical dataset.
> > >
> > > I am still not clear about the choice of normalizing flow architectures that the authors intend to consider.
> > > For 2D toy experiments the rebuttal states using GLOW, at the same time for the query regarding the applicability to complex datasets, the rebuttal states that it is beyond the scope.
> > >
> > > Moreover, wrt to the visual quality of VAEs, the rebuttal states the goal is the improvement of the ELBO and not the reconstruction, and the impact of latent dimensions on the same. Does it imply that the results depend more on the architectural choices.
> > >
> > > Given the above, I will keep my score.

---

> > > > ### Author Response · Authors · 2023-11-22
> > > >
> > > > Thank you for your response. Please let us clarify some details:
> > > >
> > > >  **"the rebuttal states using GLOW, at the same time for the query regarding the applicability to complex datasets, the rebuttal states that it is beyond the scope."**
> > > >
> > > > The Glow model, when used for *image modelling*, employs *convolutional neural networks*, and FF-kernelisation is directly applicable when *MLP neural nets* are used in a coupling layer. Therefore, we meant to say that kernelisation, where convolutional neural networks are used, requires adaptation to include parameter sharing in a convolution-like manner, which is beyond the scope of this study. Hence, to show that we CAN model complex image datasets, we opted to kernelise an architecture that exclusively relies on *MLP neural nets* for image modelling.
> > > >
> > > > **"Moreover, wrt to the visual quality of VAEs, the rebuttal states the goal is the improvement of the ELBO and not the reconstruction, and the impact of latent dimensions on the same. Does it imply that the results depend more on the architectural choices."**
> > > >
> > > > Not necessarily. Architectural choices may impact the results, but that is transferable to the kernelised variants as well. In other words, if changing the encoder and decoder architectures improves the ELBO, using flow-based priors and posterior (kernelised or non-kernelised) afterwards shall offer additional improvement in the results. This is already demonstrated in existing works that the use of flow-based priors and posteriors under the same VAE backbone helps improve the results.
> > > >
> > > > Also, we would like to point out that the choice of latent dimension is a bottleneck depending on many factors, i.e. the downstream application, and can be chosen by the applied researcher/practitioner.
> > > >
> > > > Additionally, *samples from the original data*  (**Figure 7**)  itself lacks fine details which contributed to the samples from the trained models appearing less visually appealing.
> > > >
> > > > We hope this adds clarification to our response earlier.
> > > >
> > > > Thanks!

---

> > > > > ### Comment · Reviewer_wvm3 · 2023-11-22
> > > > > **Thanks for the response**
> > > > >
> > > > > Thanks for the clarifications!
> > > > > This does clear the confusion regarding the GLOW-like vs GLOW as in Dhariwal et al.
> > > > >
> > > > > I hope these clarifications will be included in the final version.
> > > > >
> > > > > I have updated my score.

---

> > > > > > ### Author Response · Authors · 2023-11-22
> > > > > >
> > > > > > Thank you for updating the score. We will ensure the clarifications are present in the final version.

---

> ### Comment · Area_Chair_9TfK · 2023-11-20
> **Respond to authors' rebuttal**
>
> Please, confirm that you have read the author's response and the other reviewers' comments and indicate if you are willing to revise your rating.

---

### Official Review · Reviewer_ffkL · 2023-11-01

**Soundness:** 2 fair
**Presentation:** 2 fair
**Contribution:** 2 fair
**Rating:** 5
**Confidence:** 3

**Summary:**

In the paper, the authors introduce a kernelized normalizing flow paradigm that integrates kernels into the classical normalizing framework. The authors introduced theoretical fundamentals and presented results on relatively small datasets.

**Strengths:**

1. The paper introduce interesting concept in classical flow model
2. The paper has good theoretical fundaments.

**Weaknesses:**

1. In Fig 1, authors should add results from more methods, like FFJORD
2. How does the model work on a spiral 2D dataset?
3. In the main paper, we do not have any image datasets. In the appendix, we have Kuzushiji-MNIST dataset. Authors should evaluate the model on MNIST, CIFAR, and CELEBA data special when we compare methods with Glow.
4. In the paper, there is a lack of some illustration of the method. It can help to understand what exactly the kernels are in coupling layers.
5. Section 3.1 is unclear. Authors to fast introduce formula (1). Maybe some simple example to present the main idea.

**Questions:**

1. How the method works on large image datasets.
2. Ho the model works on a 2D dataset in comparison to FFJORD.

---

> ### Author Response · Authors · 2023-11-18
>
> **"In Fig 1, authors should add results from more methods, like FFJORD"**
>
> In the toy experiments(Figure 1), our focus was on illustrating the effectiveness of kernelisation under the same architecture.  With toy examples in the discontinued densities (figure 3) our objective was to showcase how composite kernels within the same architecture could introduce inductive biases; therefore we did not add any other baselines/FFJORD. Whilst we acknowledge that FFJORD may yield superior performance in certain cases, it is important to note that FFJORD represents a continuous-time normalising flow solved via neural ODEs and takes significantly longer to train as shown in Table 10. Throughout our comparisons, we primarily contrasted our approach with unkernelised versions of the same architectures, unless not feasible, as seen in the low data regime experiments where overparameterisation led to overfitting, leading us to use FFJORD(having better performance than Glow or RealNVP) for meaningful comparison.
>
> **"How does the model work on a spiral 2D dataset?"**
>
> Just like with other toy datasets, kernelisation contributes to the improved performance of the model on the spiral dataset.
> | Method      | Ours            | params |
> |-------------|-----------------|--------|
> | **Ours**    | **-3.24**      | **1.8K**|
> | NN-based      | -3.29          | 18.4K   |
>
>  We will include these results in an updated version of the manuscript.
>
> **"In the main paper, we do not have any image datasets. In the appendix, we have Kuzushiji-MNIST dataset. Authors should evaluate the model on MNIST, CIFAR, and CELEBA data special when we compare methods with Glow."**
>
>  The primary focus in this work was on making normalizing flows work well in the low-data regime, where we believe our method has significant potential and practical applications. Notably, our proposed approach serves as a direct replacement for MLP-based coupling layers, traditionally employed for non-image data. In contrast, Glow-based models use to the convolutional neural networks (CNNs) for image generation. Adapting the current method for convolutional neural networks is beyond the scope of current study and reserved for future works.
>
>  However, we showcase image generation via applicability of our kernelisation to the MixerFlow architecture, a coupling layer architecture for image generation that does not rely on convolutions for weight sharing. Our kernelised method can be directly applied to coupling layer architectures for image generation that do not rely on convolutions for weight sharing. The results of this application are presented in appendix E. It's important to note that for fair comparison, we maintained the same architecture, demonstrating that kernelisation can be successfully applied to image generation tasks with improvements over the neural-net-based architecture. Additonally, we would like to highlight that visually better image generation with flows require larger models and are typically trained on multiple GPUs. Our tabular models are trained on CPUs whereas image models are trained on a single Colab GPUs. Consequently, we have evaluated smaller models for image generation. Training bigger models with fine-grained hyperparameter tuning can help generate better visualisation, which is typically the case with flow-based image generation.

---

> > ### Comment · Reviewer_ffkL · 2023-11-20
> > **Response**
> >
> > Thanks for your answers. In my opinion, the paper is written in relation to Glow, and therefore, the experiment should also address such a model. The models should be evaluated on a larger dataset.
> > I stay with my score.

---

> > > ### Author Response · Authors · 2023-11-20
> > >
> > > Thank you for your response. We would like to gently point out that our work focuses on the kernelisation of coupling layer architectures, encompassing not only Glow but also RealNVP, VAE with flow-based prior and posterior, and MixerFlow.
> > >
> > > Additionally, we would like to clarify that we have evaluated our kernelisation on large-scale datasets(e.g. 1.6 million examples in the Power dataset) as well as small datasets (500 examples subsets or 3240 examples on the Biomarker dataset) to show the versatility and efficacy of our method.

---

> ### Comment · Area_Chair_9TfK · 2023-11-20
> **Respond to authors' rebuttal**
>
> Please, confirm that you have read the author's response and the other reviewers' comments and indicate if you are willing to revise your rating.

---

### Official Review · Reviewer_12iZ · 2023-11-08

**Soundness:** 4 excellent
**Presentation:** 4 excellent
**Contribution:** 4 excellent
**Rating:** 8
**Confidence:** 3

**Summary:**

The authors introduce their research, which aims to enhance traditional normalized flow generative models. These models typically employ neural-network-based transformations to map simple prior distributions to more complex, invertible distributions for density estimation and data generation. In their study, the authors concentrate on integrating various kernels into flow-based generative models. Their goal is to accommodate smaller datasets, reduce the number of parameters (improve parameter efficiency), and lower computational costs, all while maintaining model expressiveness. The authors propose a novel method called "Ferumal flows," which extends popular coupling layer architectures such as RealNVP (Real Non-Volume Preserving) and Glow (Generative Latent Optimization) by incorporating widely-used kernels, including the Squared Exponential and Matérn kernels.

**Strengths:**

I commend the authors for their dedicated focus on tackling the challenge of low-data scenarios in generative modeling. Furthermore, I value their efforts in quantifying the improvements brought about by their proposed architectures, particularly in terms of reducing computational demands for hyperparameter tuning and training convergence compared to other approaches.

**Weaknesses:**

It would have been beneficial if the authors had provided links to a repository containing their code and models for improved accessibility and reproducibility. Additionally, given the authors' initial reference to the potential application of their models in the medical field, it would have been valuable to include an evaluation of their models' performance on a medical dataset to demonstrate their practical applicability and potential benefits in that specific domain.

**Questions:**

In your introduction, you alluded to the potential application of your models in the medical field. Why did you choose not to evaluate your models on a medical dataset as part of your research?

Many researchers and practitioners might find value in accessing your code and models for further exploration and application. Could you please share your considerations regarding the possibility of providing links or access to your code and models in the future, and if so, where could interested parties expect to find them?

---

> ### Author Response · Authors · 2023-11-18
>
> **"It would have been beneficial if the authors had provided links to a repository containing their code and models for improved accessibility and reproducibility."**
>
>  We currently include the code for our implementation in the supplementary material. Additionally, upon the public release of the paper, we will make the code readily available through a public repository for accessibility and reproducibility.
>
>  **"it would have been valuable to include an evaluation of their models' performance on a medical dataset to demonstrate their practical applicability and potential benefits in that specific domain."**
>
> We have now incorporated an evaluation of our method's performance on a medical dataset, demonstrating its practical applicability in a real life application. Please refer to Section D of the appendix for the results..

---

> ### Comment · Area_Chair_9TfK · 2023-11-20
> **Respond to authors' rebuttal**
>
> Please, confirm that you have read the author's response and the other reviewers' comments and indicate if you are willing to revise your rating.

---

> > ### Comment · Reviewer_12iZ · 2023-11-22
> >
> > I would like to thank the authors for taking the time to run experiments using the UKBiobank's biomarker data. I stay with my score.

---

### Author Response · Authors · 2023-11-18

We would like to thank all the reviewers for their valuable feedback and insightful comments. We truly appreciate the time and effort you put into reviewing our work. We are happy that the reviewers found that our work "well written", "interesting", and "theoretically well founded". Based on the valuable suggestions of the reviewers, we have now added a medical dataset in our analysis and an image generation experiment.

We defer to the individual responses for more details.

---

### Meta-Review · Area_Chair_9TfK · 2023-12-08

**Metareview:**

Summary:

Normalising flows are probabilistic machine learning models that are capable of jointly solving the task of density estimation and generative modelling (sampling). Unfortunately, since they are parameterised in terms of a pushforward and their log-likelihood is determined using a change-of-variables formula involving a Jacobian, the mappings involved must be invertible, and are often even more constrained (e.g. through coupling layers). Invertibility imposes a big constraint on the model, meaning that the only lever one may pull to obtain expressiveness is often the depth and overparameterisation of the neural network. This results in data-hungriness, and makes them highly unsuitable for modelling tabular and low-dimensional data. In their study, the authors concentrate on integrating various kernels into flow-based generative models. Their goal is to accommodate smaller datasets, reduce the number of parameters (improve parameter efficiency), and lower computational costs, all while maintaining model expressiveness. The authors propose a novel method called "Ferumal flows," which extends popular coupling layer architectures such as RealNVP (Real Non-Volume Preserving) and Glow (Generative Latent Optimization) by incorporating widely-used kernels, including the Squared Exponential and Matérn kernels. The authors introduce a kernelised version of normalising flows, which are suitable for modelling low-dimensional and tabular data. Efficacy is demonstrated on a set of benchmark datasets.

Strengths:

- The paper ideas presented in this paper are conceptually simple and do not require a strong leap-of-faith on the reader's end.
- The mathematical idea (a representer theorem) appears to be mostly sound.
- The experiments on tabular / low dimensional data are convincing and demonstrate the appeal of the method. A suitable benchmark is considered.
- The paper is well-written. Text, equations, tables and figures are appropriately laid out and given the right amount of real-estate.
- The paper introduce interesting concept in classical flow model
- The paper has good theoretical fundaments.
- The proposed approach is novel and interesting.
- The paper correctly claims that normalizing flows which employ neural network based coupling layers are data and parameter hungry. The proposed kernel based approach in contrast is data and parameter efficient.
- The results in Table 3 show that the proposed approach shines in the low data regime. It clearly outperforms FFJORD and obtains impressive results even when only 500 data samples are available.
- The paper is well written and theoretically well founded.

Weaknesses:

- It would have been valuable to include an evaluation of their models' performance on a medical dataset to demonstrate their practical applicability and potential benefits in that specific domain.
- In the main paper, we do not have any image datasets. In the appendix, we have Kuzushiji-MNIST dataset. Authors should evaluate the model on MNIST, CIFAR, and CELEBA data special when we compare methods with Glow.
- In the paper, there is a lack of some illustration of the method.
- Methods like FFJORD (\cf Figure 2 in FFJORD) report better results compared to the proposed approach (as shown in Table 1). The performance advantage of FFJORD is even more apparent in case of the challenging discontinuous checkerboard dataset in Figure 3 (supplementary). It is not clear if the proposed model has the modelling capacity to capture complex distributions.
- The proposed method is outperformed significantly by FFJORD, although FFJORD uses more parameters as reported in Table 4.
- The method is motivated by alluding to medical settings in Section 1 and 5.4, where data availability is often limited. However, the method is never applied to any medical data.
- It not clear if the proposed model can be applied to complex data distributions such as images.
- The qualitative examples of samples generated by the proposed model trained on the Kuzushiji-MNIST dataset as shown in Figure 4-6 are not promising.

Recommendation:

A majority of reviewers vote for acceptance. I, therefore, recommend accepting the paper and encourage the authors to use the feedback provided to improve the paper for the camera ready version.

**Justification For Why Not Higher Score:**

Reviewers point out some weaknesses and one leads slightly towards rejection.

**Justification For Why Not Lower Score:**

A majority of reviewers vote for acceptance. The only reviewer voting for rejection submitted a rather short review of lower quality.

---

### Decision · Program_Chairs · 2024-01-16

Accept (poster)